# Electric Field-Induced Settling and Flotation of Flocs in Mixed Aqueous Suspensions of Poly(methyl methacrylate) and Aluminosilicate Hollow Particles

**DOI:** 10.3390/ma18061289

**Published:** 2025-03-14

**Authors:** Hiroshi Kimura, Mirei Sakakibara

**Affiliations:** Department of Chemistry and Biomolecular Science, Faculty of Engineering, Gifu University, Gifu 501-1193, Japan; sakakibara.mirei.b8@s.gifu-u.ac.jp

**Keywords:** aqueous colloidal dispersion, electric field, hollow particles, co-flocculation, floc settling, floc flotation

## Abstract

When a horizontal electric field is applied, the sedimentation velocity of particles increases, a phenomenon known as Electrically Induced Rapid Separation (ERS). Hollow particles with a lower density than water exhibit an increased flotation velocity under an electric field. This study investigates the ERS effect in mixed suspensions containing particles denser than water and hollow particles with lower density. In the absence of an electric field, the denser particles settle while the hollow particles float, and their behavior is independent of the ratio of hollow particles to the total number of particles (*α*). However, when a DC electric field of 0.4 V/mm is applied, the behavior becomes dependent on *α*. For *α* < ~0.90, all particles sediment, whereas for *α* > ~0.93, all particles float. This suggests that the electric field induces a co-floc formation between the denser and hollow particles. Additionally, for the first time, a co-floc formation under an electric field was directly observed using a digital microscope. By adjusting *α* and applying an electric field, it is possible to control the sedimentation, flotation, or stabilization of the particle system. This study provides new insights into electric field-assisted particle separation and highlights its potential applications in colloidal science and materials science.

## 1. Introduction

Colloidal particles in polar liquids, such as water, maintain a stable dispersion state due to the presence of an electrical double layer. However, as particle size increases and the influence of the Brownian motion diminishes, dispersion stability is compromised, making the particles more susceptible to sedimentation or flotation due to density differences with the dispersing medium. When a horizontal electric field is applied to such a colloidal suspension in water, the sedimentation velocity of the particles increases. This phenomenon is referred to as the Electrically Induced Rapid Separation (ERS) effect by the authors. It is evident that the ERS effect is highly likely caused by floc formation induced by the applied electric field [1,2,3,4]. The ERS effect has been confirmed in aqueous dispersions of various colloidal particles, including poly(methyl methacrylate) (PMMA) particles, silica particles, bentonite (a clay mineral), and others. Therefore, it is expected that the ERS effect will also manifest in dispersions with a wide range of different dispersed materials. It is well known that when an external electric field is applied to colloidal particles in water, interfacial electrokinetic phenomena such as electrophoresis occur. However, despite its significance, the phenomenon of particle aggregation and redispersion induced by the application and removal of an electric field has received little attention. Notably, while numerous studies have been conducted on the behavior of colloidal particles in highly confined spaces, such as on electrode surfaces, as discussed later, investigations into the broader effects of electric field-induced aggregation and redispersion in bulk dispersions remain largely unexplored.

The intriguing aggregation behavior of colloidal particles, such as polystyrene particles (diameter: 2.5–10 μm), under a vertically applied electric field has been investigated when the particles are sedimented on a horizontal electrode surface [5]. This study suggests that the interaction between the electrical double layer on the particle surface and the external electric field induces electroosmotic flow, which in turn draws colloidal particles together, leading to aggregation. Furthermore, Kim et al. [6] observed the motion of two particles on an electrode under an AC electric field, reporting that particles are attracted to each other at low frequencies (below 500 Hz) but separate at higher frequencies. Similarly, Wei et al. [7] investigated the irreversible aggregation of polystyrene particles under an AC electric field (1.5 kHz frequency, 1.3 V voltage, 50 μm electrode spacing). Their findings indicated that the aggregation process follows two distinct stages: Reaction-Limited Aggregation (RLA) and Diffusion-Limited Aggregation (DLA). Fraden et al. [8] reported that polystyrene particles form chain-like structures under an AC electric field (30–150 kHz). Under high-intensity electric fields (1000 V/cm), the chains remained highly stable as long as the field was applied. In contrast, under low-intensity electric fields, the chains exhibited a dynamic equilibrium state, where formation and dissociation occurred repeatedly due to their shorter lifespan. Several other notable studies have explored the behavior of colloidal particles in confined spaces, particularly on electrode surfaces. Examples include electrophoretic deposition techniques [9,10], colloidal crystallization on electrodes [11,12,13], and particle assembly control using rotating electrodes [14]. In contrast, the colloidal behavior investigated in this study occurs not in a confined two-dimensional space (e.g., on an electrode) but in a three-dimensional environment. Kim et al. [15] proposed a novel numerical simulation method based on the Smooth Profile (SP) method to accurately and efficiently analyze electrophoretic phenomena in charged colloidal dispersions. Their study discusses the effects of an electric field on the electrical double layer and dispersion state of colloidal particles, particularly focusing on particle electrophoretic mobility, deformation, and overlap of electrical double layers in dense dispersions.

Our research group’s previous studies have revealed that the sedimentation velocity of poly(methyl methacrylate) (PMMA) particles and montmorillonite in water increases significantly when a horizontal DC electric field of several V/mm is applied. In cases where the particle volume fraction was at most 0.001 and the electric field strength reached up to 1.0 V/mm DC, the sedimentation velocity increased by up to several hundred times [1]. These findings strongly suggest that particle flocculation occurs under a horizontal electric field. Furthermore, the ERS effect is highly dependent on the direction of the applied electric field. Kimura and Tsuchida [2] investigated the application of a vertical DC electric field to a PMMA particle aqueous dispersion. When an upward electric field was applied, the negatively charged PMMA particles exhibited an increase in sedimentation velocity simply due to the additive effect of electrophoresis directing them toward the bottom of the cell. This observation indicates that flocculation did not occur. In contrast, under a downward electric field, pronounced convection was observed within the cell, maintaining a relatively uniform dispersion state. After an extended period, some PMMA particles adhered to the upper positive electrode, while the remaining particles sedimented at the lower negative electrode. As a result, rapid sedimentation via the flocculation of PMMA spheres was not observed under a vertical electric field. These findings highlight the critical role of electric field direction in the manifestation of the ERS effect, emphasizing that the orientation of the applied field significantly influences particle behavior.

Recently, Kimura [16] demonstrated that aluminosilicate-based hollow particles exhibit rapid ascent when a horizontal electric field is applied to their aqueous suspension. This finding supports the idea that hollow particles undergo floc formation under an electric field. A key question that arises from this observation is the following: How will a mixed system behave when combining particles denser than water (previously studied) with hollow particles, which have a lower density than water, under an applied electric field? To address this question, the present study aims to elucidate the influence of hollow particle mixing on the ERS effect. We investigated the ERS effect using a horizontal electric field applied to a mixed aqueous suspension of PMMA particles and hollow particles, a system previously employed by the authors in ERS effect studies.

## 2. Experimental

### 2.1. Sample Preparation

The monodisperse poly(methyl methacrylate) (PMMA) particles were purchased from Sekisui Plastics Co., Ltd. (Osaka, Japan). The particle diameter *d*_PMMA_ was 5.3 ± 0.4 μm (Figure 1a), with a density of 1.20 g/cm^3^. The zeta potential *ζ* was measured using a Zetasizer Nano ZS (Malvern Instruments Ltd., Malvern, UK) and determined to be −40 mV [15]. Aluminosilicate hollow particles were provided by Taiheiyo Cement Co. (Tokyo, Japan). The particle shells are composed of aluminosilicate glass. These particles exhibit a relatively polydisperse distribution, ranging from 1.0 μm to 8.0 μm, with an average particle diameter *d*_hollow_ of approximately 1.4 μm (Figure 1b). As determined from transmittance measurements, the density of the hollow particles was found to be 0.77 g/cm^3^. The zeta potential was measured as −31 mV. PMMA particles and hollow particles were each dispersed in ultrapure water (Milli-Q Advantage A10, Millipore Co., Burlington, MA, USA), followed by desalination treatment using ion-exchange resin (AG501-X8 (D), Bio-Rad Lab., Inc., Hercules, CA, USA). The deionization process was carried out for over two months for PMMA particles and over eight months for hollow particles. The total volume of the dispersion injected into the measurement cell was 3.5 mL for all samples. The volume fraction *ϕ* and particle number *N* of PMMA particles and hollow particles in the dispersion are listed in Table 1. The fraction of hollow particles relative to the total number of particles, denoted as *α*, was defined as follows:(1)$$\alpha =\frac{N_{hollow}}{N_{PMMA}+N_{hollow}}$$
where *N*_PMMA_ represents the number of PMMA particles in the dispersion and *N*_hollow_ represents the number of hollow particles. Here, *α* = 0 corresponds to a pure PMMA particle suspension, while *α* = 1 corresponds to a pure hollow particle suspension.

### 2.2. Measurement Method

A rectangular transparent cell (10 mm × 10 mm × 45 mm) was used for the experiment. Inside the cell, Type 304 stainless steel plates were set parallel to each other along the cell walls, serving as electrodes (Figure 1c). The gap between the electrodes was 9.8 mm. After injecting the aqueous dispersion into the cell, the top of the cell was sealed with a rubber cap. The dispersion was gently stirred and placed on a horizontal measurement stage. Within two minutes of setting up the cell, video recording commenced. Simultaneously with video recording, a DC electric field was applied to the dispersions using a synthesized function generator (FG110, Yokogawa Test & Measurement Co., Tokyo, Japan). Based on the authors’ previous experiments, the electric field strength *E* was fixed at 0.4 V/mm DC, a condition that suppresses electrolysis while ensuring the manifestation of the ERS effect. The flotation velocity of hollow particles in the absence of an electric field was evaluated using transmittance measurements conducted with a custom-built apparatus. A laser beam (wavelength: 632.8 nm, beam diameter: 0.5 mm) was directed perpendicular to the cell walls and perpendicular to the electric field direction. The observation point was positioned at the center between the electrodes, at a height *H*_obs_ = 5 mm from the bottom of the cell. Within two minutes of setting up the cell, transmittance measurements were initiated. Simultaneously, a fixed-point observation camera was used for low-speed imaging of the dispersion state (frame interval: 10 s; frame rate: 10 FPS), capturing the particle distribution along the same direction as the laser beam irradiation. The dispersion near the surface, which contained floated flocs after 12 h of electric field application, was collected and observed using a digital microscope (BA81-6T-1080M, Shimadzu RIKA Co., Tokyo, Japan). All measurements were conducted at a temperature of 25 °C.

## 3. Results and Discussion

### 3.1. Potential Energy Curves from DLVO (Derjaguin–Landau–Verwey–Overbeek) Theory

To evaluate the dispersion stability of PMMA particles and hollow particles in water, DLVO potential energy curves [17,18] were constructed for each particle system (Figure 2). The potential energy is expressed as follows:

(2)$$V=V_{R}+V_{A}=\pi d\varepsilon {\Psi_{0}}^{2}\;\exp(-\frac{H}{L_{D}})-\frac{A_{H}d}{24H}$$
where *V* represents the total potential energy, *V*_R_ is the electrostatic potential energy, *V*_A_ is the van der Waals attractive potential energy, *d* is the particle diameter, *ε* is the dielectric constant of water, *Ψ*_0_ is the surface potential of the particle, *H* is the surface separation, *L*_D_ is the Debye screening length, and *A*_H_ is the Hamaker constant. The formula for *V*_R_ is valid when *d* ≫ *L*_D_ [19]. The *A*_H_ value used for PMMA particles was 6.3 × 10^−20^ J [20]. Since the Hamaker constant for hollow particles is unknown, we assumed values in the range of 0.5 × 10^−20^ J to 3.0 × 10^−20^ J and plotted the corresponding potential energy curves. According to the report by Bergström [21], the Hamaker constants of silica (*A*_H_ = 0.63 × 10^−20^ J) and aluminum oxide (*A*_H_ = 2.97 × 10^−20^ J) in water are presented. Assuming that only 1:1-type electrolytes are present in the aqueous medium, the *L*_D_ value can be determined using the following equation [22]:(3)$$L_{D}=\frac{1}{\kappa }=\sqrt{\frac{\varepsilon k_{B}T}{2000e^{2}cN_{A}}}$$

Here, *κ* is the Debye–Hückel parameter, *k*_B_ is the Boltzmann constant (1.38 × 10^−23^ J/K), *T* is the absolute temperature, *e* is the elementary charge, *c* is the electrolyte concentration, and *N*_A_ is Avogadro’s number (6.02 × 10^23^ mol⁻¹). At 25 °C, *L*_D_ (in meters) for an electrolyte concentration *c* (in mol/L) can be approximated as follows:(4)$$L_{D}=\frac{3.04}{\sqrt{c}}\times 10^{-10}$$

Based on Kimura’s previous study [23], the electrolyte concentration was estimated as *c* = 1.0 × 10^−5^ mol/L, resulting in *L*_D_ = 96 nm. In Figure 2, the vertical axis represents the total potential energy normalized by thermal energy, indicating how much larger the interaction energy between particles is compared to thermal motion. A potential barrier of 25 or higher prevents particle aggregation upon collision [24]. These results confirm that both PMMA particles and hollow particles exhibit sufficiently strong electrostatic repulsion, preventing aggregation. Furthermore, when negatively charged PMMA particles and hollow particles are mixed, their interaction is unlikely to lead to floc formation, even upon close approach.

### 3.2. Time Evolution of Dispersion State Under Field-Free and Electric Field Conditions

The particle Reynolds number *Re*_p_ is given by the following equation [25]:(5)Rep=ρwvdη
where *ρ*_w_ represents the density of water, *v* is the flotation velocity of the hollow particles, *d* is the diameter of the hollow particles, and *η* is the viscosity of water. Using these parameters, *Re*_p_ for hollow particles was calculated to be 4.4 × 10^−7^. Since this value is extremely small, it confirms that the Stokes approximation is valid for describing the flotation velocity of the hollow particles. The flotation velocity of the hollow particles was determined from the transmittance change in the dispersion at *H*_obs_ = 5 mm (Figure 3). The transmittance exhibited a linear increase immediately after the start of measurement, eventually reaching a steady-state value. By applying Stokes’ law, the density of the hollow particles was estimated based on the elapsed time at the intersection of the steady-state transmittance and the initial slope of the transmittance curve. As a result, the density of the hollow particles was determined to be 0.77 g/cm^3^.

The time-dependent dispersion state of the PMMA and hollow particle-mixed aqueous suspensions was recorded under field-free conditions (Figure 4). Representative images for selected *α* values are shown, while images for other *α* values are provided in Appendix A. In the PMMA aqueous suspension (*α* = 0), the PMMA particles sedimented over time, as observed in Figure 4(a1–a9). In the *α* = 0.50 mixed dispersion (Figure 4(b1–b8)), PMMA sedimentation was confirmed, but the ascent of hollow particles was not clearly visible to the naked eye. This is likely due to the extremely low volume fraction of hollow particles (1.3 × 10^−6^ at *α* = 0.50). For *α* = 0.97 (Figure 4(c1–c8)), both the sedimentation of PMMA particles and the ascent of hollow particles were observed. In the hollow particle aqueous suspension (*α* = 1), hollow particles were confirmed to ascend over time (Figure 4(d1–d10)). To the right side of each image set, the changes in dispersion state over time are indicated, specifically sedimentation (SD) and ascent (ASC). When no clear change in the dispersion state was observed, it was labeled as n/o (not observed). These results demonstrate that, under field-free conditions, PMMA and hollow particles sediment and ascend independently, confirming that no significant interaction occurs between the two particle types. The Péclet number *Pe* is expressed as(6)$$\mathrm{Pe}=\frac{\pi d^{4}\Delta \rho g}{12k_{B}T}$$
where Δ*ρ* represents the density difference between the particles and water. The *Pe* values for PMMA particles and hollow particles were 99 and 0.55, respectively. Compared to the PMMA aqueous suspension, the less distinct interface observed in the hollow particle aqueous suspension can be attributed to the greater contribution of diffusion, as well as the relatively broad particle size distribution of the hollow particles.

The time-dependent dispersion state of the mixed suspension under an applied electric field (*E* = 0.4 V/mm DC) was recorded (Figure 5). Representative images for selected *α* values are shown, while images for other *α* values are provided in Appendix A. In the PMMA aqueous suspension (*α* = 0), a highly intriguing phenomenon was observed (Figure 5(a1–a10)). After the application of the electric field, the particles migrated toward the anode due to electrophoresis, and approximately 600 s later, the aggregated particles near the anode rapidly sedimented. In our previous ERS effect studies, the typical particle volume fraction used was 0.0005. Under the same electric field strength (*E* = 0.4 V/mm DC), such large-scale electrophoresis had never been observed. In those cases, only a small fraction of particles near the cathode migrated toward the anode, with a maximum displacement of approximately 1 mm. As a result, the colloid-rich region maintained a nearly “horizontal” interface with the water phase while undergoing rapid sedimentation. The large-scale electrophoresis observed in this study prior to the manifestation of the ERS effect suggests an important finding: a certain minimum volume fraction (*ϕ* > ~0.0001) is required for the ERS effect to occur. In the *α* = 0.50 mixed dispersion (Figure 5(b1–b10)), the dispersion state of hollow particles remained indistinct, similar to the field-free condition. However, PMMA particles exhibited significantly accelerated sedimentation compared to the field-free condition, and electrophoresis was again observed. For *α* = 0.97 (Figure 5(c1–c10)), electrophoresis was observed, followed by the unexpected flotation of all particle groups. This was a highly surprising result. In the hollow particle aqueous suspension (*α* = 1), the ERS effect of hollow particles (rapid flotation under an applied electric field) was reconfirmed (Figure 5(d1–d10)). In this case, electrophoresis was barely observed. Since the volume fraction of hollow particles was *ϕ* = 0.0005, the “horizontal” interface between the colloid-rich and water phases ascended, consistent with previous observations. These findings suggest that the dispersion state of the mixed suspension under an applied electric field depends on the mixing ratio of PMMA and hollow particles.

### 3.3. Influence of Hollow Particle Mixing Ratio on the ERS Effect

To clarify the effect of the mixing ratio on the ERS effect, we compared the dispersion states at a fixed time point (*t* = 1400 s) after the start of measurement under both field-free conditions and with an applied electric field (*E* = 0.4 V/mm DC) (Figure 6). Under field-free conditions (Figure 6(a1–a9)), no significant sedimentation of PMMA particles or flotation of hollow particles was observed at this time. The subsequent changes in the dispersion state are indicated below each image. As can be seen, the changes in dispersion state under field-free conditions were consistent with expectations, consisting of PMMA particle sedimentation and hollow particle flotation. Specifically, in the range of *α* = 0–0.97, PMMA sedimentation was observed, while in the range of *α* = 0.90–1, hollow particle flotation was observed. As mentioned earlier, for *α* = 0.10 and 0.50, the volume fractions of hollow particles were *ϕ* = 1.5 × 10^−7^ and *ϕ* = 1.3 × 10^−6^, respectively, which were too small for their flotation to be visually confirmed. On the other hand, with an applied electric field (Figure 6(b1–b9)), it became evident that the particle ratio strongly influenced whether the entire particle system underwent sedimentation or flotation. In the range of *α* = 0–0.90, the entire particle system sedimented, whereas for *α* > 0.92, the entire system floated. In the range of *α* = 0.90–0.95, electrohydrodynamic (EHD) convection was observed. These results suggest that floc formation between PMMA and hollow particles, as well as complex interfacial electrokinetic phenomena such as electrophoresis and electroosmotic flow, play a significant role in determining the dispersion behavior.

These results strongly suggest that PMMA particles and hollow particles form co-flocs. The occurrence of the ERS effect requires the presence of an electrical double layer [4]. It is most reasonable to consider that the electrical double layers formed around the negatively charged PMMA particles and hollow particles in water are distorted in the direction of the applied electric field and that the induced dipole moments attract each other, leading to the formation of co-flocs. The simplest hypothesis is that, under an applied electric field, a single PMMA particle associates with multiple hollow particles to form a single floc. To examine this further, the relationship between the number ratio of hollow particles to a single PMMA particle, denoted as *β*, and the fraction of hollow particles, *α*, was investigated (Figure 7). *β* can be expressed in terms of *α* as follows, based on Equation (1).(7)$$\beta =\frac{N_{hollow}}{N_{PMMA}}=\frac{\alpha }{\left(1-\alpha \right)},\;\mathrm{where}\;\alpha \;\neq 1$$

The results show that as *α* increases, *β* also increases, exhibiting a sharp rise for *α* > 0.90. When a single PMMA particle forms a floc together with *β* hollow particles, the density of the floc, *ρ*_floc_, can be determined using the following equation.(8)$$\rho_{floc}=\frac{\rho_{PMMA}\cdot V_{PMMA}+\beta \cdot \rho_{hollow}\cdot V_{hollow}}{V_{PMMA}+\beta \cdot V_{hollow}}$$
where *ρ*_PMMA_ and *ρ*_hollow_ represent the densities of the PMMA particle and the hollow particle, respectively. Similarly, *V*_PMMA_ and *V*_hollow_ denote the volumes of a single PMMA particle and a single hollow particle, respectively. If a single PMMA particle forms a co-floc with a large number of hollow particles, the density of the co-floc becomes equal to the density of water (1.0 g/cm^3^) when *β* = 48, meaning a particle number ratio of 1:48 (*α*~0.98). However, in reality, the complete flotation of all particles under the electric field was observed at approximately *α* = 0.92. When *α* = 0.92, the corresponding *β* value is 11.5. If a single PMMA particle (diameter: 5.3 μm, density: 1.20 g/cm^3^) forms a floc with 11.5 hollow particles (diameter: 1.4 μm, density: 0.77 g/cm^3^), a straightforward calculation—where the floc density is determined by dividing the total mass of the floc by its total volume (Equation (8))—yields a floc density of 1.13 g/cm^3^. This value exceeds 1.0 g/cm^3^, indicating that the PMMA particles should all settle. However, in reality, a single PMMA particle does not necessarily form a floc with many hollow particles. Instead, multiple PMMA particles may be included in a single floc, or some PMMA particles may form flocs with only a few hollow particles. In such cases, flocs with *α* values lower than approximately 0.98 may settle completely at the bottom of the cell (as shown in Figure 8). This would significantly increase the proportion of hollow particles remaining in the dispersion, making it highly likely that the *α* value of the flocs formed by the remaining PMMA and hollow particles exceeds 0.98. Since hollow particles, either in isolation or within hollow particle-only flocs, always ascend, the increase in the volume fraction of hollow particles in the upper region of the dispersion over time supports this explanation. The validity of this hypothesis will be further examined in future studies, which will investigate the effects of total particle volume fraction and applied electric field strength on the ERS effect, as well as conduct detailed analyses of the composition of sedimented and floated particle groups induced by the ERS effect. A mixed suspension with *α* = 0.97 was subjected to microscopic observation after being left undisturbed for 12 h under both field-free conditions and an applied electric field (*E* = 0.4 V/mm DC). The dispersion near the surface, containing floated flocs, was collected and observed (Figure 9). Under field-free conditions (Figure 9(a1,a2)), only hollow particles were observed near the surface, indicating that PMMA particles had sedimented and were absent from this region. In contrast, when an electric field was applied (Figure 9(b1,b2)), both PMMA and hollow particles were detected near the surface. Since the observed PMMA particles were not forming flocs, it suggests that they had redispersed after the removal of the electric field. These results strongly support the hypothesis that PMMA and hollow particles form co-flocs under an applied electric field.

## 4. Conclusions

Colloidal particles in water form flocs and exhibit the ERS effect under a DC electric field of a few V/mm or less. In this study, the ERS effect was investigated for a system in which hollow particles, which have a lower density than water, were mixed with PMMA particles. In the single-component system, the sedimentation velocity of PMMA particles increased under an applied electric field, while the flotation velocity of hollow particles also increased. In the mixed particle system, when the fraction of hollow particles, *α*, was approximately 0.90 or lower, the entire particle system sedimented, whereas for *α* values of approximately 0.93 or higher, all particles floated. These results indicate that PMMA and hollow particles form co-flocs under an applied electric field. These results were directly observed using a digital microscope. The findings of this study suggest the possibility of controlling the entire particle system’s sedimentation, flotation, or stabilization at arbitrary velocities by applying an electric field to a mixture of particles with different densities. To further investigate the observed convection inside the cell and regulate its intensity, future experiments under an alternating electric field are also desirable.

## Figures and Tables

**Figure 1 materials-18-01289-f001:**
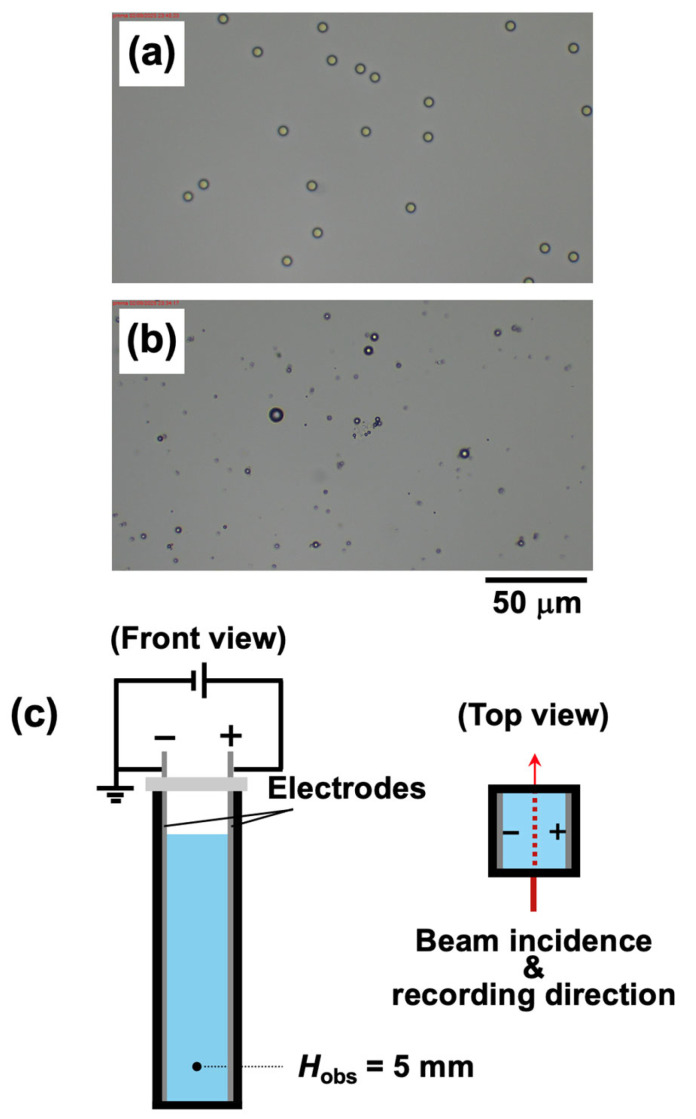
(**a**) PMMA particles, (**b**) hollow particles, and (**c**) schematic diagram of the experimental cell used for imaging the dispersion and transmittance measurements. (**a**,**b**) The particle volume fraction of both PMMA and hollow particles was *ϕ* = 0.0005. About 5 μL was dropped onto a glass slide and covered with a cover glass. The objective lens magnification was 40×.

**Figure 2 materials-18-01289-f002:**
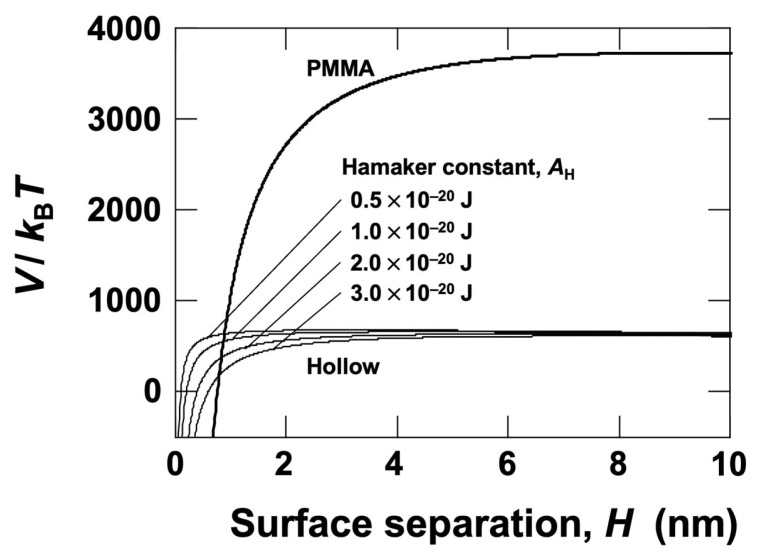
DLVO potential energy curves for PMMA particles and hollow particles in deionized water. The *A*_H_ values for hollow particles are plotted in the range of 0.5 × 10^−20^ J to 3.0 × 10^−20^ J, while the *A*_H_ value for PMMA particles is 6.3 × 10^−20^ J. Parameters: *d*_PMMA_ = 5.3 μm; *d*_hollow_ = 1.4 μm; *ζ*_PMMA_ = −40 mV; *ζ*_hollow_ = −31 mV; *L*_D_ = 96 nm; *T* = 298 K.

**Figure 3 materials-18-01289-f003:**
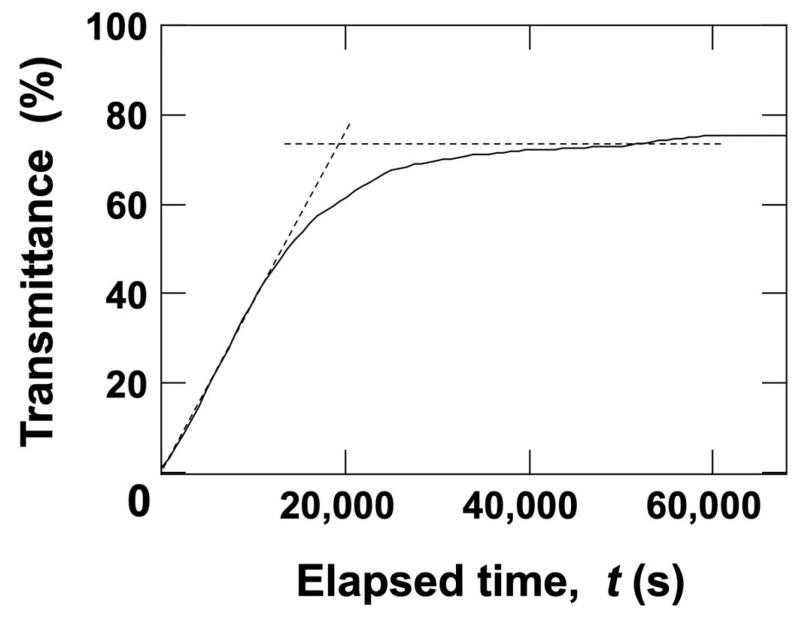
Time-dependent transmittance of the hollow particle aqueous dispersion at 25 °C. *ϕ* = 0.0005; *H*_obs_ = 5 mm.

**Figure 4 materials-18-01289-f004:**
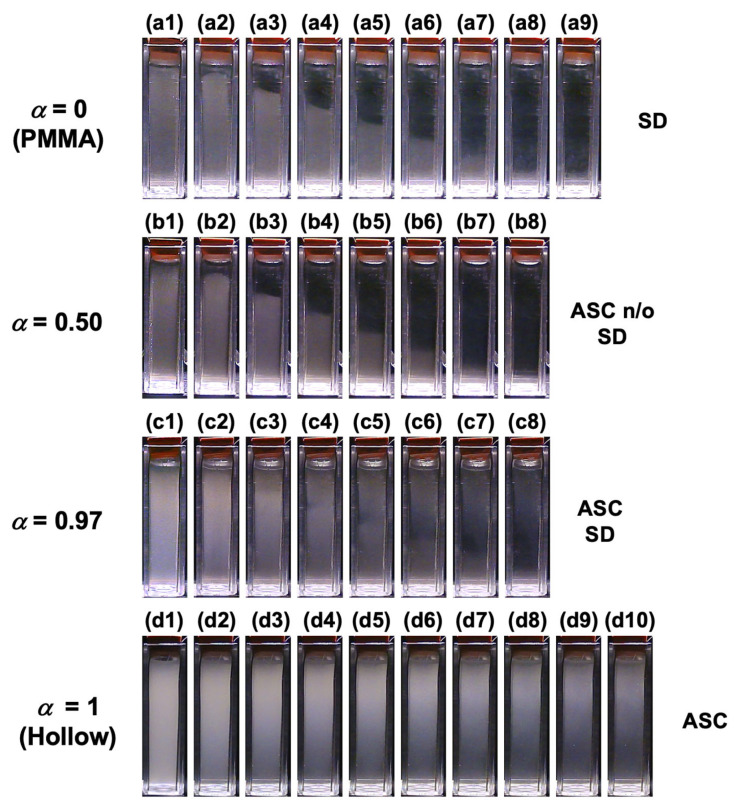
The time-dependent dispersion state of the mixed aqueous suspension of hollow and PMMA particles at 25 °C under field-free conditions. (**a1**–**a9**) *α* = 0 (PMMA), (**b1**–**b8**) *α* = 0.50, (**c1**–**c8**) *α* = 0.97, (**d1**–**d10**) *α* = 1 (Hollow). (**a1**,**b1**,**c1**,**d1**) *t* = 0, (**a2**,**b2**,**c2**,**d2**) *t* = 2000 s, (**a3**,**b3**,**c3**,**d3**) *t* = 4000 s, (**a4**,**b4**,**c4**,**d4**) *t* = 6000 s, (**a5**,**b5**,**c5**,**d5**) *t* = 8000 s, (**a6**,**b6**,**c6**,**d6**) *t* = 10,000 s, (**a7**,**b7**,**c7**,**d7**) *t* = 12,000 s, (**a8**,**b8**,**c8**,**d8**) *t* = 14,000 s, (**a9**,**d9**) *t* = 16,000 s, (**d10**) *t* = 18,000 s.

**Figure 5 materials-18-01289-f005:**
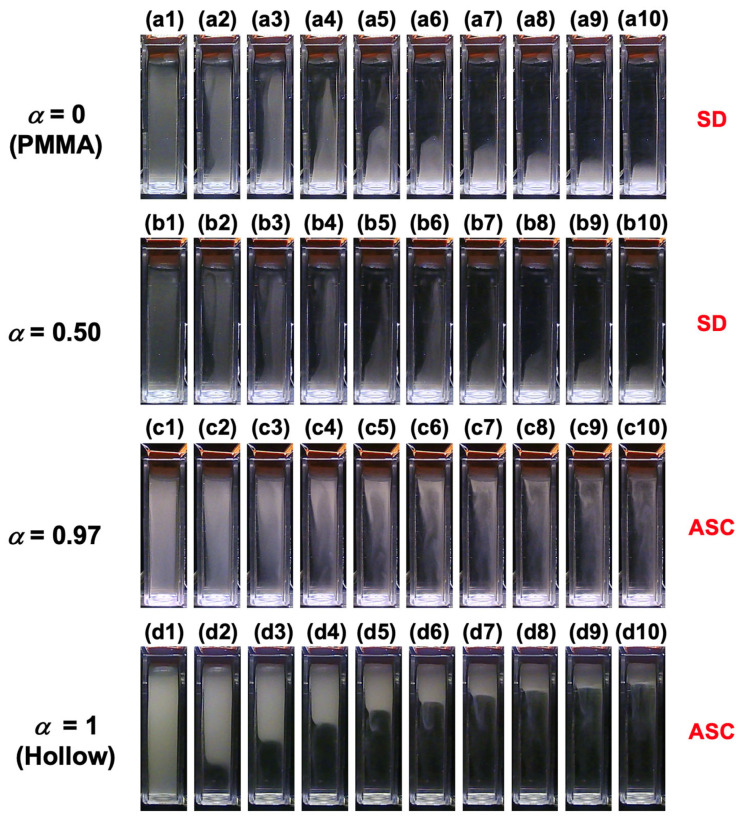
The time-dependent dispersion state of the mixed aqueous suspension of hollow and PMMA particles at 25 °C under an applied electric field (*E* = 0.4 V/mm DC). (**a1**–**a10**) *α* = 0 (PMMA), (**b1**–**b10**) *α* = 0.50, (**c1**–**c10**) *α* = 0.97, (**d1**–**d10**) *α* = 1 (Hollow). (**a1**,**b1**,**c1**,**d1**) *t* = 0, (**a2**,**b2**,**c2**,**d2**) *t* = 200 s, (**a3**,**b3**,**c3**,**d3**) *t* = 400 s, (**a4**,**b4**,**c4**,**d4**) *t* = 600 s, (**a5**,**b5**,**c5**,**d5**) *t* = 800 s, (**a6**,**b6**,**c6**,**d6**) *t* = 1000 s, (**a7**,**b7**,**c7**,**d7**) *t* = 1200 s, (**a8**,**b8**,**c8**,**d8**) *t* = 1400 s, (**a9**,**b9**,**c9**,**d9**) *t* = 1600 s, (**a10**,**b10**,**c10**,**d10**) *t* = 1800 s.

**Figure 6 materials-18-01289-f006:**
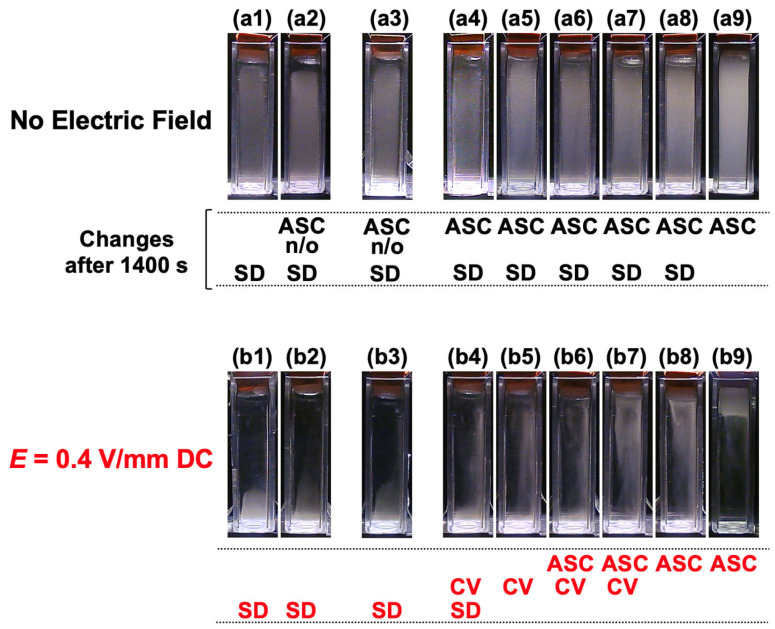
The dispersion state of the mixed aqueous suspension of hollow and PMMA particles at 25 °C after 1400 s. (**a1**–**a9**) without an electric field, (**b1**–**b9**) *E* = 0.4 V/mm DC. (**a1**,**b1**) *α* = 0 (PMMA), (**a2**,**b2**) *α* = 0.10, (**a3**,**b3**) *α* = 0.50, (**a4**,**b4**) *α* = 0.90, (**a5**,**b5**) *α* = 0.92, (**a6**,**b6**) *α* = 0.93, (**a7**,**b7**) *α* = 0.95, (**a8**,**b8**) *α* = 0.97, (**a9**,**b9**) *α* = 1 (Hollow). Major particle movements under *E* = 0.4 V/mm DC: ascent (ASC), convection (CV), sedimentation (SD).

**Figure 7 materials-18-01289-f007:**
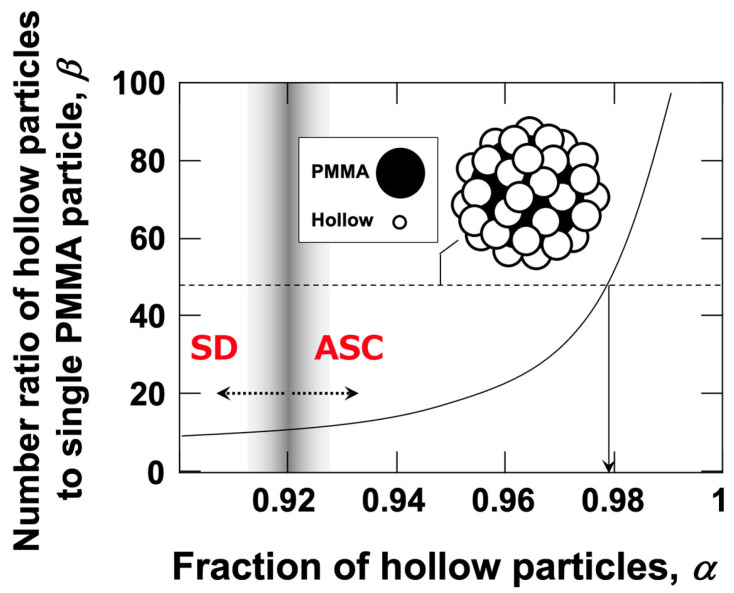
Effect of the mixing ratio *α* on the number of hollow particles per single PMMA particle at 25 °C. Schematic representation of floc formation under an applied electric field: when one PMMA particle is associated with 48 hollow particles, the floc density becomes 1.0 g/cm^3^.

**Figure 8 materials-18-01289-f008:**
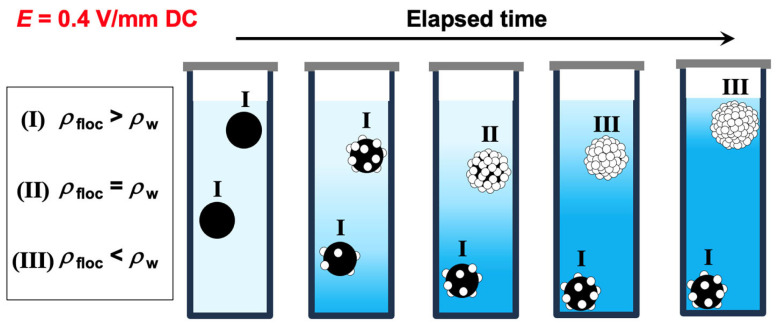
Schematic illustration of the time-dependent flotation (upper floc) and sedimentation (lower floc) of co-flocs formed by PMMA and hollow particles under an applied electric field. The floc density *ρ*_floc_ is classified as (I) greater than the water density *ρ*_w_, (II) equal to *ρ*_w_, and (III) smaller than *ρ*_w_. Hollow particles are represented in white, with their flotation-induced distribution changes depicted using a gradient color representing the dispersion medium. Hollow particles forming flocs with PMMA particles are shown as white circles.

**Figure 9 materials-18-01289-f009:**
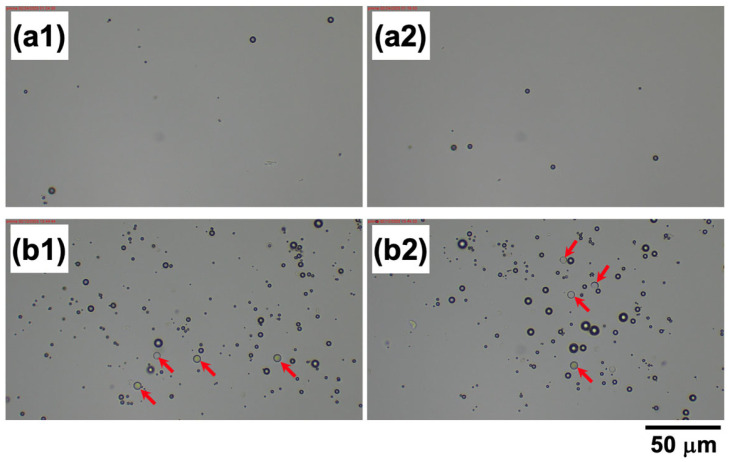
Digital microscope images of the surface region containing floated flocs in the mixed suspension with *α* = 0.97 after being left undisturbed for 12 h at 25 °C. (**a1**,**a2**) without an electric field, (**b1**,**b2**) with an applied electric field (*E* = 0.4 V/mm DC). Here, (**a1**,**a2**) correspond to different observed regions, as do (**b1**,**b2**). The red arrows indicate PMMA particles.

**Table 1 materials-18-01289-t001:** Volume fraction *ϕ* and particle number *N* of PMMA and hollow particles corresponding to the fraction of hollow particles, *α*, in the total particle count.

*α*	*ϕ* _PMMA_	*ϕ* _hollow_	*ϕ* _total_	*N* _PMMA_	*N* _hollow_	*N* _total_
0	7.1 × 10^−5^	0	7.1 × 10^−5^	3.2 × 10^6^	0	3.2 × 10^6^
0.10	7.1 × 10^−5^	1.5 × 10^−7^	7.2 × 10^−5^	3.2 × 10^6^	3.6 × 10^5^	3.6 × 10^6^
0.50	7.1 × 10^−5^	1.3 × 10^−6^	7.3 × 10^−5^	3.2 × 10^6^	3.2 × 10^6^	6.4 × 10^6^
0.90	7.1 × 10^−5^	1.2 × 10^−5^	8.3 × 10^−5^	3.2 × 10^6^	2.9 × 10^7^	3.2 × 10^7^
0.92	7.1 × 10^−5^	1.5 × 10^−5^	8.7 × 10^−5^	3.2 × 10^6^	3.7 × 10^7^	4.0 × 10^7^
0.93	7.1 × 10^−5^	1.7 × 10^−5^	8.9 × 10^−5^	3.2 × 10^6^	4.3 × 10^7^	4.6 × 10^7^
0.95	7.1 × 10^−5^	2.5 × 10^−5^	9.6 × 10^−5^	3.2 × 10^6^	6.1 × 10^7^	6.4 × 10^7^
0.97	7.1 × 10^−5^	4.3 × 10^−5^	1.1 × 10^−4^	3.2 × 10^6^	1.0 × 10^8^	1.1 × 10^8^
1	0	5.0 × 10^−4^	5.0 × 10^−4^	0	1.2 × 10^9^	1.2 × 10^9^

## Data Availability

The original contributions presented in this study are included in the article. Further inquiries can be directed to the corresponding author.

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
