# Peer review of "Electric Field-Induced Settling and Flotation of Flocs in Mixed Aqueous Suspensions of Poly(methyl methacrylate) and Aluminosilicate Hollow Particles"

_materials, 2025, doi:10.3390/ma18061289_

Round 1
Reviewer 1 Report
Comments and Suggestions for Authors
The study provides valuable insights into the Electrically Induced Rapid Separation effect in mixed aqueous suspensions of PMMA and hollow particles. By varying the fraction of hollow particles and applying an electric field, the study effectively analyzes its impact on particle behaviour. The manuscript is well-structured and clearly written. However, some minor modifications could further enhance the clarity and depth of the presentation.
- To improve the clarity and coherence of the manuscript, Figure 2 should be relocated to immediately follow the first paragraph of Section 3.1. Figures should be placed as close as possible to their first mention in the text to enhance readability
- The Hamaker constant for hollow particles is assumed to lie within the range of 0.5 × 10⁻²⁰ J to 3.0 × 10⁻²⁰ However, the basis behind this selection should be provided, either through relevant literature or experimental estimations, to support the validity of this range.
- To improve transparency and allow readers to verify the calculations more easily, please specify the exact numerical values of Avogadro’s number and the Boltzmann constant in the manuscript.
- A more comprehensive explanation of the underlying forces leading to co-floc formation would greatly benefit the manuscript.
- The manuscript states that complete flotation should occur at 𝛼 = 0.98, but in experiments, this is observed at 𝛼 = 0.92. The proposed explanation of preferential sedimentation of PMMA particles is insufficiently detailed. It would be helpful to clarify which specific forces or interactions lead to this deviation from the theoretical predictions.
Author Response
Response to Reviewer 1
Dear Reviewer,
We sincerely appreciate your valuable comments and suggestions, which have significantly contributed to improving our manuscript. Below, we provide our responses to your comments and explain the corresponding revisions made to the manuscript.
- Figure 2 Placement
Comment: To improve the clarity and coherence of the manuscript, Figure 2 should be relocated to immediately follow the first paragraph of Section 3.1. Figures should be placed as close as possible to their first mention in the text to enhance readability.
Response: We have relocated Figure 2 near the beginning of Section 3.1. to enhance readability and maintain coherence in the manuscript.
- Rationale for the Assumed Range of the Hamaker Constant
Comment: The Hamaker constant for hollow particles is assumed to lie within the range of 0.5 × 10–20J to 3.0 × 10–20 However, the basis behind this selection should be provided, either through relevant literature or experimental estimations, to support the validity of this range.
Response: As reported by Bergström (1997), the Hamaker constants for silica (AH = 0.63 × 10–20 J) and aluminum oxide (AH = 2.97 × 10–20 J) in water have been documented. Based on these values, we adopted the range of 0.5 × 10–20 J to 3.0 × 10–20 J for our hollow particles. We have included this explanation in Lines 185–187 and cited Bergström (1997) as Reference [21] in the revised manuscript.
- Enhancing Transparency in Calculations
Comment: To improve transparency and allow readers to verify the calculations more easily, please specify the exact numerical values of Avogadro’s number and the Boltzmann constant in the manuscript.
Response: We have explicitly added Avogadro's number (6.02 × 1023 mol–1) and Boltzmann’s constant (1.38 × 10–23 J/K) in Lines 193–195 to improve the transparency of our calculations.
- Mechanism Behind Co-floc Formation
Comment: A more comprehensive explanation of the underlying forces leading to co-floc formation would greatly benefit the manuscript.
Response: The formation of co-flocs under an applied electric field is primarily attributed to the distortion of the electrical double layers surrounding the negatively charged PMMA and hollow particles, leading to induced dipole moments that cause mutual attraction. This explanation has been added in Lines 326–330.
- Discrepancy Between Theoretical and Experimental Flotation Conditions
Comment: The manuscript states that complete flotation should occur at ? = 0.98, but in experiments, this is observed at ? = 0.92. The proposed explanation of preferential sedimentation of PMMA particles is insufficiently detailed. It would be helpful to clarify which specific forces or interactions lead to this deviation from the theoretical predictions.
Response: We have thoroughly revised Section 3.3. to clarify the differences between theoretical and experimental results, rewriting the explanation to ensure greater clarity. Additionally, Equations 7 and 8 have been introduced to support this discussion. These changes are reflected in Lines 334–365.
Once again, we appreciate the constructive feedback and believe that the revisions have improved the manuscript. We look forward to any further comments you may have.
Best regards,
Hiroshi Kimura
On behalf of all authors
Reviewer 2 Report
Comments and Suggestions for Authors
Comment to the authors
I proceeded to analyze the manuscript entitled:
Electric Field-Induced Settling and Flotation of Flocs in Mixed Aqueous Suspensions of Poly(methyl methacrylate) and Aluminosilicate Hollow Particles,
written by
Hiroshi Kimura and Mirei Sakakibara
This manuscript explores the Electrically Induced Rapid Separation (ERS) effect in mixed suspensions of dense and hollow particles. The authors found, after and experimental investogation, that under a 0.4 V/mm DC field, particle behavior depends on 𝛼: for 𝛼 < ~0.90, all particles settle, while for 𝛼 > ~0.93, all float. This indicates electric field-induced co-floc formation, confirmed via digital microscopy by authors, being the first time of such reporting. The authors claim that by adjusting 𝛼 and applying an electric field, sedimentation, flotation, or stabilization of the system can be controlled.
The topic is, in my opinion, interesting and of interest for scholars working on colloids. The figures are suggestive and support the statements.
References are in small amount but they are in decent amount considering that the field they are writing about is quite narrow itself and reveal that the authors are well aware of what was written in the field. The work reported here presents the result of the continuation of their work.
The article is well written, using good English, in my opinion, but a careful check would improve it.
Yet I found aspects that, in my opinion, raise minimum concerns and require improvement and additional clarification, as indicated on each item, and they are mentioned below.
-Line 164 : “DLVO” define it before its first use; check the rest of the manuscript in his respect.
-Line 332: “At ≈ 0.92, simple calculations indicate that the density of co-flocs should not exceed that of water”. Show the calculation, to make things clear.
Author Response
Response to Reviewer 2
Dear Reviewer,
We sincerely appreciate your detailed review and valuable suggestions, which have helped us to improve the clarity and quality of our manuscript. Below are our responses to your comments along with the corresponding revisions made to the manuscript.
- Definition of DLVO Theory (Line 164)
Comment: Line 164: “DLVO” define it before its first use; check the rest of the manuscript in his respect.
Response: We have added the full form of DLVO theory as "Derjaguin–Landau–Verwey–Overbeek theory" at its first mention (Line 166) to ensure clarity for all readers. Furthermore, we have reviewed the manuscript to confirm that all abbreviations are appropriately defined upon their first occurrence.
- Theoretical Justification for Flotation at ? ≈ 0.92 (Line 332)
Comment: Line 332: “At a ≈ 0.92, simple calculations indicate that the density of co-flocs should not exceed that of water”. Show the calculation, to make things clear.
Response: We have carefully revised Section 3.3. to clarify the discrepancy between experimental and theoretical results. Additionally, we have introduced Equations 7 and 8, along with a detailed explanation of the parameters used, to provide a clear theoretical foundation for the observed flotation behavior (Lines 334–365).
We are grateful for your constructive feedback, which has helped us refine our work. We believe that the revisions have enhanced the manuscript and look forward to any further comments you may have.
Best regards,
Hiroshi Kimura
On behalf of all authors
Reviewer 3 Report
Comments and Suggestions for Authors
The paper submitted by Kimura and Sakakibara investigates the effect of an electric field on the aqueous suspensions of poly(methyl methacrylate) and aluminosilicate hollow particles. The manuscript is clear, well written and the conclusions are supported by the results. The following comments must be addressed:
- fig 1: how the photos a and b were obtained? The caption must be completed.
- can the particle's sizes be determined by DLS?
- the Turbiscan technology would have been very useful for this study.
- the conclusions of the study can be applied to other types of polymer particles?
Author Response
Response to Reviewer 3
Dear Reviewer,
We sincerely appreciate your valuable comments and constructive suggestions, which have greatly contributed to improving our manuscript. Below, we provide our responses and outline the corresponding revisions made to the manuscript.
- Additional Details in Figure 1 Caption
Comment: fig 1: how the photos a and b were obtained? The caption must be completed.
Response: We have included additional details regarding the imaging conditions in the caption of Figure 1 (Lines 134–136). Specifically, additional details have been provided regarding the particle volume fraction, droplet volume, and objective lens magnification. The main text (Section 2.2.) mentions that the dispersion near the surface, including the floc that floated after 12 hours of electric field application, was collected, and that a digital microscope (BA81-6T-1080M, Shimadzu RIKA Co., Tokyo, Japan) was used (Lines 160–164).
- Possibility of Particle Size Measurement Using DLS (Dynamic Light Scattering)
Comment: can the particle's sizes be determined by DLS?
Response: In this study, we used monodisperse PMMA particles and polydisperse hollow particles. We have chosen to focus on the average particle sizes to simplify the discussion as much as possible. Additionally, due to convection effects inside the measurement cell under the applied electric field, sedimentation and flotation velocities cannot be quantitatively analyzed at this stage. For these reasons, we have not included DLS measurements in this study.
- Applicability of Turbiscan Technology
Comment: the Turbiscan technology would have been very useful for this study.
Response: Thank you for this valuable suggestion. At present, our laboratory does not have access to the Turbiscan system, so we are unable to conduct these measurements immediately. However, we recognize that this method could provide a quantitative evaluation of the dispersion state. If we acquire the necessary equipment in the future, we would be very interested in applying this technique. Nevertheless, as mentioned in the manuscript, convection within the cell may pose a challenge for quantitative analysis.
- Applicability of Findings to Other Polymer Particles
Comment: the conclusions of the study can be applied to other types of polymer particles?
Response: Since polymer particles such as PS and PU generally acquire charges in water, it is highly likely that the ERS effect and the electric field-induced dispersion control observed in this study would also apply to such systems. We have concisely described this point in the Conclusion section (Lines 410–413).
We sincerely appreciate your insightful feedback, which has helped improve the manuscript. We believe these revisions have strengthened our work, and we welcome any further comments you may have.
Best regards,
Hiroshi Kimura
On behalf of all authors